# Atmospheric Propagation Studies and Development of New Instrumentation for Astronomy, Radar, and Telecommunication Applications in the Subterahertz Frequency Range

Yurii Balega [1], Gregory Bubnov [2], Mikhail Glyavin [2], Aleksandra Gunbina [2], Dmitry Danilevsky [3], Grigory Denisov [2], Andrey Khudchenko [4], Ilya Lesnov [2], Andrey Marukhno [1,2], Kirill Mineev [2], Sergey Samsonov [2], Gennady Shanin [3] and Vyacheslav Vdovin [2,4,*]

1 Special Astrophysical Observatory of the Russian Academy of Sciences, 369167 Karachai-Cherkessian, Russia; balega@sao.ru (Y.B.); mas@sao.ru (A.M.)
2 Institute of Applied Physics of the Russian Academy of Sciences, 603950 Nizhny Novgorod, Russia; bubnov@ieee.com (G.B.); glyavin@ipfran.ru (M.G.); gunbina@ipfran.ru (A.G.); den@ipfran.ru (G.D.); lesnov@ipfran.ru (I.L.); mineevkv@gmail.com (K.M.); samsonov@ipfran.ru (S.S.)
3 RT-70 Observatory, Tashkent 100000, Uzbekistan; danilevsky_d@mail.ru (D.D.); suffa@mail.ru (G.S.)
4 Astro Space Center Lebedev Physical Institute of the Russian Academy of Sciences, 119991 Moscow, Russia; khudchenko@asc.rssi.ru
* Correspondence: vdovin@ipfran.ru

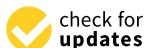



**Featured Application: Applications of subTHz extremely-high-power gyrotrons and low-noise receivers for new antenna projects in astronomy, space radar, and telecommunication have been analyzed.**

**Abstract:** Current progress in the development of new subterahertz instruments discussed in this paper, including antennas, high-power gyrotrons, and low-noise receivers, provides a wide range of possible applications. Atmospheric absorption has now become a major limitation in the application of such high-performance subTHz systems, and the choice of the optimal location of antennas for astronomical, radar, and communication systems is important. The latest results of studying the microwave astroclimate in northern Eurasia are presented. Based on these studies, new perspectives and corrected plans for installing new antennas at the Suffa Plateau and in the Caucasus are formulated, and possible applications of new instruments such as radars for locating space debris and communication hubs for deep space communications, which are based on extremely-high-power gyrotrons and low-noise superconducting receivers, are discussed.

**Keywords:** Terahertz waves; MM and subMM waves; radars; space communications; atmosphere propagation; astroclimat; radio telescopes; gyrotrons; receivers

## 1. Introduction

The subterahertz (subTHz) range today is not a fully established term among engineers and researchers who develop technologies for receiving and transmitting radio waves and operate antenna equipment at frequencies from 0.1 to 1 THz, covering, in now more familiar terms, the short-wave part of the millimeter range (wavelengths of 3–1 mm) and the long-wave part of the submillimeter range (1–0.3 mm). A unifying feature of the subTHz range is a specific combination of electronic and photonic technologies used, as well as the approximate uniformity of the conditions for propagation of these waves in the atmosphere, which has a set of transparency windows separated by high-level absorption lines of atmospheric gases (mainly $O_2$ and $H_2O$) [1]. For radio waves that are longer than subTHz, the well-developed technologies of classical semiconductor radio-electronics are used, and their propagation is under the relatively weak influence of the atmosphere over very long distances. Using a more conventional term, "THz (terahertz)", for this range is not entirely correct because it includes waves shorter than 0.3 mm (higher than 1 THz).

Waves shorter than the subTHz ones (shorter than 0.3 mm or over 1 THz) are mainly based on the photonic technologies, and up to optical range in its near-IR part, they have practically no transparency windows when operating on the Earth at the sea level and slightly higher, except only extremely exotic cases like the high Atacama Plateau [2]. In recent decades, the subTHz range has been a most actively and dynamically developing research and development area.

The subTHz range promises and has already given unprecedented breakthroughs in astrophysics: specifically, the network of a dozen subTHz antennas included in the Event Horizon Telescope (EHT) [3] has made it possible to see the shadow of a black hole and identify different types of black holes.

In telecommunications, subTHz radiation will provide both deep space communication channels of unprecedented capacity [4], which ensure a data transfer rate being quite close to that of optical channels and give new prospects for the terabit capacity of terrestrial mobile communication at 6G and beyond [5].

There are important tasks for subTHz radar, especially in the context of sounding space debris and other space objects, including extremely small ones [6,7].

Certainly, very good progress has been achieved in the development of contemporary tools for astronomy, space communications, and radars in the optical frequency range. Technologies of laser generation and CCD receivers using the optical wavelengths are more impressive and open up the unique possibilities for astronomy, communications, and radars. Examples of remarkable developments are described in a lot of publications, and some of them are commented upon below.

When preparing this article, we were primarily motivated by the progress of the technologies that have been developed in the last decade for generating and receiving subTHz radiation, and the current progress in the worldwide development of subTHz antennas and the 70 m RT-70 antenna of the Suffa observatory, in particular.

However, optical methods do not resolve all problems, and different spectral windows are needed for the operation of radars, telescopes, and communication channels in the atmosphere. In addition to purely spectral features that make it possible to see new colors of the spectrum, the subTHz range has other advantages.

Despite the significant atmospheric influence on subTHz waves, in some cases, radars and telecommunications have serious advantages over optical means, which are potentially better than subTHz ones, both in terms of capacity and resolution, but are "blind" in sunlight or other lighting conditions and completely unsuitable for operation in cloudy conditions. SubTHz waves have advantages for radar and communication applications compared with low-frequency (CM and DM) radio waves [6–8]. They are, e.g., high precision, i.e., a favorable ratio of the antenna diameter and the beamwidth [9],

$$\theta \sim \frac{\lambda}{D} \qquad (1)$$

and have high range resolution for a fixed relative bandwidth:

$$\Delta r \sim \frac{c}{2B} \qquad (2)$$

The HUSIR radar [7] has been designed for geostationary satellite imaging with a 3 cm range resolution. The Space Debris Surveillance Radar with the 0.01 m$^2$ resolution has a detection range of ~1500 km, providing an angular precision of 10 inches. Unfortunately, the HUSIR project was frozen several years ago, but its key components have been already manufactured by Communications & Power Industries, Palo Alto, CA, USA [10]. Similar systems with close parameters can be developed for the Suffa Observatory [11] by the team [12] using the subTHz components presented here and developed in collaboration with the Institute of Applied Physics (IAP RAS) and GYCOM. The main instrument of the Suffa Observatory, the RT-70 antenna, is a Cassegrain structure with a parabolic primary mirror and a hyperbolic secondary mirror. Further calculations have been made for this

antenna. However, the USA and Russia, which have been mentioned above, are not the only leaders in R&D of extremely high-power subTHz sources. Some rather effective groups working in Europe [13], Japan [14], and China have already developed several systems that meet the competition.

The next section of the article is devoted to the progress achieved mainly by the authors of this paper together and in collaboration with leading international groups. The progress in recent years has been so significant that the factors dominating now among the limitations imposed on the use of subTHz technologies for the development of radio astronomy, radar, and telecommunications, are not technical restrictions, but natural ones, namely, the atmosphere.

Certainly, we should make a note of recent achievements of the photonic technologies in those applications too. Remarkable results have been obtained not only for terrestrial applications, but also for space-based ones. Impressive developments of optical technologies [15], including those used in communications [16] and radar applications [17] are quite broad and undoubtedly have significant advantages over longer-wavelength subTHz systems, but their limitations are also known. The applications of subTHz systems presented here are useful additions to optical systems, not only for astronomy, where we have additional spectral windows due to their usage, but for communications and radar applications as well. The third section of this article is devoted to the results of recent studies of the conditions for propagation of the subTHz waves at the most significant and promising sites in northeastern Eurasia conducted by authors during the last decade for the development and installation of new antenna systems for subTHz radioastronomy, radar, and telecommunications. Other, rather exotic, applications proposed for the subTHz range will be only mentioned in this paper, rather than presented in detail, since the authors are not experts in the related field, but are only aware of these research areas.

## 2. Methods and Hardware for Application of subTHz Instruments

### 2.1. Gyrotrons: High-Power subTHz Sources and Amplifiers

Here, we consider approaches to the problem of propagation of high-power microwave beams using examples of currently available powerful and ultra-powerful (up to 1 MW) pulsed or continuous-wave (CW) subTHz radiation sources, which propagate in the subTHz atmospheric transparency window of ~1.3 mm or ~240 GHz. A line of very powerful subTHz generators and amplifiers has been developed and manufactured in collaboration with IAP RAS and GYCOM [18–22]. For such powerful amplifiers and generators, efficiency is an important factor. The efficiency of gyro-devices (including those based on gyro-frequency harmonics) can be increased significantly (by about a factor of one and a half) through the use of recuperation schemes, which has been demonstrated repeatedly in experiments [22].

This progress in developing unique generators, which generate powers of up to the megawatt level, and receivers [23] in the subTHz range with the noise close to the fundamental quantum limit expands the range of possible subTHz applications radically.

Applications of powerful microwave sources (see Figure 1) actively developed over the recent years in plasma heating installations, future thermonuclear power plants, and accelerators [18], as well as when fulfilling several scientific and engineering tasks ranging from high-resolution laboratory spectroscopy [19] to drilling wells for geothermal energy plants and SETI problems [24], do not involve some exotic cases of channeling subTHz radiation through the atmosphere [25,26] and, therefore, go beyond the scope of this paper. The influence of the atmosphere is essential for us in the following subTHz applications considered here.

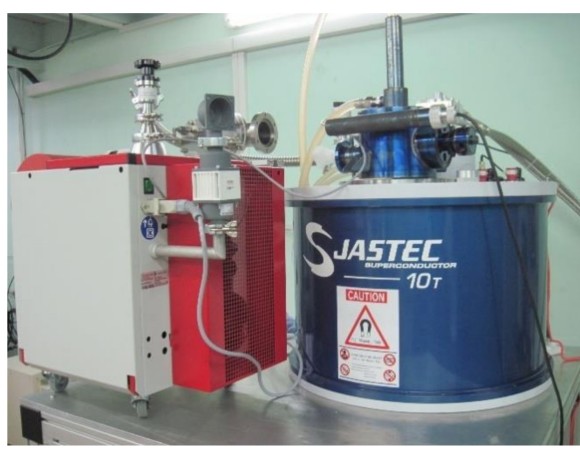

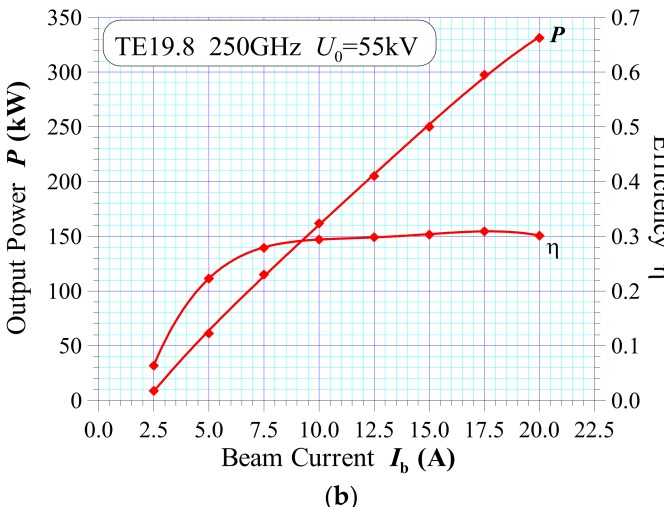

(**a**)

(**b**)

**Figure 1.** (**a**) Gyrotron for 250 GHz (330 kW with pulse duration about 100 microseconds, or 250 kW with CW) and (**b**) its output power and efficiency.

1. Use of radar installations for identifying the positions of space debris, the volume of which has been dramatically increasing in the near-Earth orbits in recent years, and also for scanning the lunar and planetary surfaces [27,28] from the Earth (ground-based low-power radar may have some other applications in addition to the above).

2. Telecommunications. Interplanetary and deep-space missions, like flights to Mars and Venus, and the L2 missions, require high-performance space communications [4,28], which will be bound to use the THz range. Here, some old projects of sending messages from the Earth to extraterrestrial civilizations come to one's mind [24], but even short-term prospects of a new generation of terrestrial mobile communications, such as 7G, imply the use of THz waves: 7G is expected to distribute signals between mobile consumers at frequencies of about 0.7 THz [5], while the exchange between cells will be organized via fiber-optic links.

3. There are currently more exotic projects of using powerful microwave beams sent through the atmosphere not as information carriers, but as energy transport means, as contrasted with the first two cases. The examples that have appeared over recent years in the THz community include power supplied from the Earth to a spacecraft for propulsion [25] and power supplied from near-Earth solar power stations to terrestrial consumers via powerful microwave beams [26].

The signal-to-noise ratio (SNR) is the key characteristic for telecommunication and, even more so, radar tasks. The use of signal sources of extremely high power and effective ultra-high-sensitivity receivers, the technology of which has also made good progress over the last decade [23], makes it possible to provide extremely high SMR levels.

The best performance parameters of the gyrotrons developed by the authors, primarily for nuclear fusion, but proposed here for other applications, are: 250 GHz/330 KW in the pulsed regime, 263 GHz/1 KW in the pulsed and continuous-wave (CW) regimes, and 527 GHz/250 W in the pulsed and CW regimes. It is definitely better than the proposals of the competitors listed here [13,14].

The sensitivity of heterodyne receivers [23], which have been developed and used in applications by leading international groups, including the authors, has reached 1–2 quantum limits in the subTHz range, and the sensitivity of the developed subTHz superconducting detectors [29] cooled down to 0.1 K is essentially determined by the background level only.

### 2.2. Receivers, Antennas, and the Atmosphere as Critical Factors of subTHz Systems

The parameters of the presented subTHz generators, amplifiers (see Figure 2), and receivers developed by the team of the coauthors are sufficiently high to ensure the proper

technical signal/noise ratio when such equipment will be used with new antenna systems for any applications mentioned in the Introduction. However, the above-stated atmosphere issue persists. The main task now is to study the effect of the atmospheric conditions on the subTHz radiation propagating through the atmosphere. To find a solution for the highly efficient channeling of ultra-powerful microwave beams through the atmosphere, we used the techniques and tools developed in astronomy to study the propagation of microwave signals of astrophysical objects through the Earth's atmosphere. Here, we regard the atmosphere as an inhomogeneous dispersing and dissipative medium. The inhomogeneity is presented through one of the generally accepted flat-layered models, within the framework of which the extensive statistics of atmospheric characteristics collected at various locations and altitudes above the sea level have been processed.

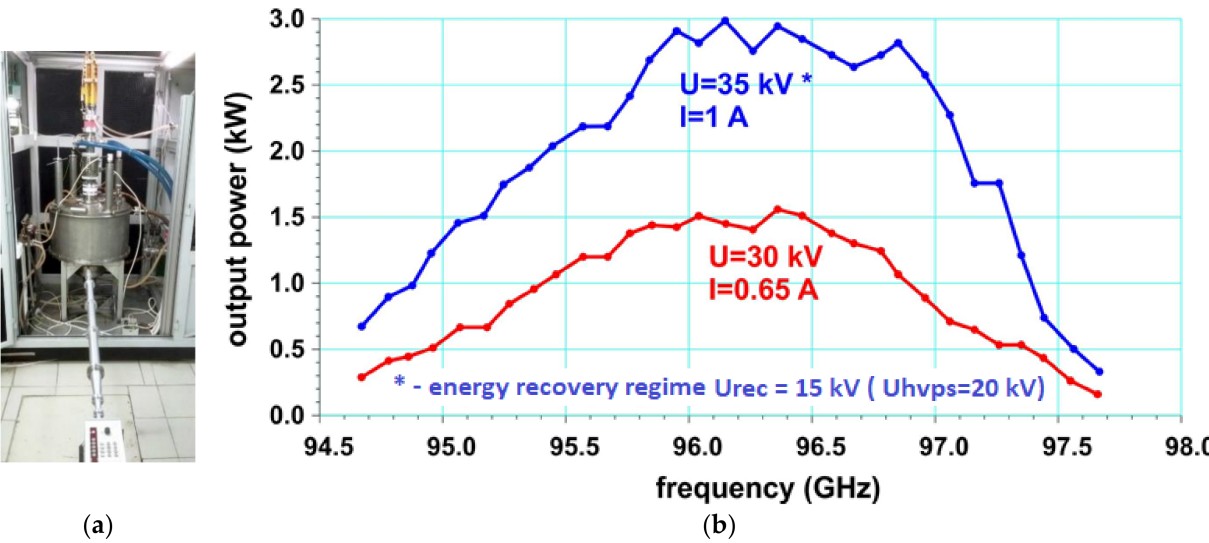

(**a**)  (**b**)

**Figure 2.** (**a**) GYRO amplifier for ~100 GHz, 3 KW (gyro TWT), and (**b**) its specifications.

Speaking about the noticeable progress in the development of large subTHz antenna systems [2,11,30], powerful gyro transmitters (megawatt sources with a line width of 1 Hz and amplifiers with a band of 2 GHz) [12,18], and high-sensitivity quantum level receivers [23,28], we can have a new look at the prospects opening up in applications of the subTHz range, now taking the atmosphere effect into account. In this connection, the choice of the optimal location of the antenna for astronomy, satellite communications, and especially for subTHz radars (due to the two-way passing of radar beams through the atmosphere) is quite critical.

A two-band MIAP-2 radiometer [31] was developed to study the atmospheric absorption of electromagnetic waves in the subTHz frequency range by the method of direct subTHz measurement (see Figure 3). The database publicly available on the ZENODO web page [32] is the result of more than ten years of measurements at various geolocations and different altitudes above the sea level, from coastal sites to altitudes of ~3 km in the vicinity of the Suffa Plateau and the North Caucasus. It is complemented here by the measurements made in summer 2021 at the Elbrus (5500 m) [33] and Mount Aktashtau (3200 m), which is close to the Suffa Plateau.

The direct measurements fulfilled within the framework of the presented research efforts at specifically selected sites showed a radical difference (by an order of magnitude or greater) in the propagation conditions of subTHz beams depending on the location of the antenna and seasonal and other variations, which must be taken into account when considering a choice of the location for antenna systems.

At present, extensive satellite and meteorological data are available, from which certain conclusions can be drawn about the prospects of using subTHz instruments without visiting the actual installation site, but based on the data obtained by various methods and

tools in multiple ranges, in particular, using the optical or even GPS and V-band equipment indicating the water vapor quantity. This handy tool for preliminary estimates was used by us for the initial assessment of test locations and the choice of the directions for the expeditions presented here.

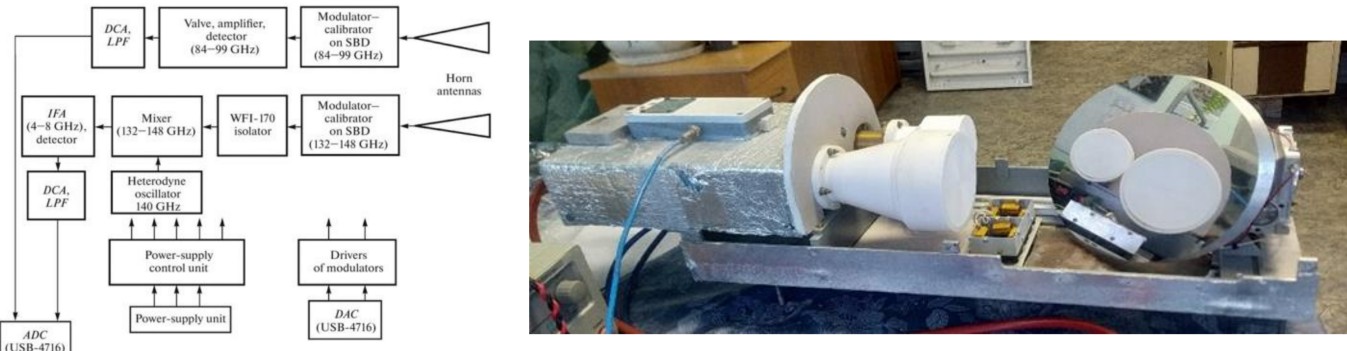

**Figure 3.** Block diagram and photo of MIAP-2, a sub-THz radiometric system for atmospheric measurements [31].

However, not only water and vapor absorption, but also other atmospheric components are essential for microwave astroclimate since they have almost no effect on the propagation of beams in the optical or long-wavelength radio bands (e.g., GPS and CM wave radiometers). For example, the 3 mm transparency window is limited at the edges by two lines of oxygen absorption, mainly transparent to optical and longer radio waves. Moreover, the satellite data based on relatively wide beams (up to tens of kilometers) do not consider the natural ruggedness of mountainous surfaces. These data are suitable for the case of flat plateaus like Pamir and Tibet [34] and have no significance for rugged areas with very high surface RMC. In the analysis of the Suffa Plateau in the search for promising sites for the EHT network expansion [28,35], this error was very well demonstrated. The integral altitude in [3] was averaged at 2000 m, which is significantly lower than the natural (2400 m) elevation of the RT-70 telescope being under construction now. The resulting analysis shows that the atmospheric absorption is higher than expected. At the same time, both theory and experiments show a significant decrease in atmospheric absorption with growing altitude. The satellite data [3] estimate the water contribution from the surface based on ~1 square degree assessed as the integrated result of the peaks and ravines, which is like making the rugged surface flat. However, for rough surfaces, more detailed information is required. For example, at the Suffa site, the difference in altitudes inside the square degree was more than 700 m. For a reasonable selection of sites for subTHz instruments, long-term direct monitoring of the atmosphere at the base site, and direct measurements of the wavelengths at which the designed tools are supposed to operate, are required.

There is a particular interest in the research performed on the Suffa Plateau and in the North Caucasus due to the projects to deploy two VLBI (very large base interferometer) antennas of the subTHz frequency range under the international Suffa project [11]. Within the Suffa project, as usually is the case, some space debris radar surveillance is to be completed along with astronomical observations. In addition, such large antennas, along with astronomical research, are generally used for distant space communications [4].

A typical profile of atmospheric absorption in the subTHz range is shown in Figure 4. Certain implementations of astronomy, communication, and radar applications in the subTHz range are available in atmospheric transparency windows only. Figure 4 is not presenting rainfall and snowfall effects. The reason is that the places chosen for the subTHz antenna are very dry without essential rains and thick clouds, and occasionally (a few percent of the time), when there is precipitation and thick clouds, no work is in progress.

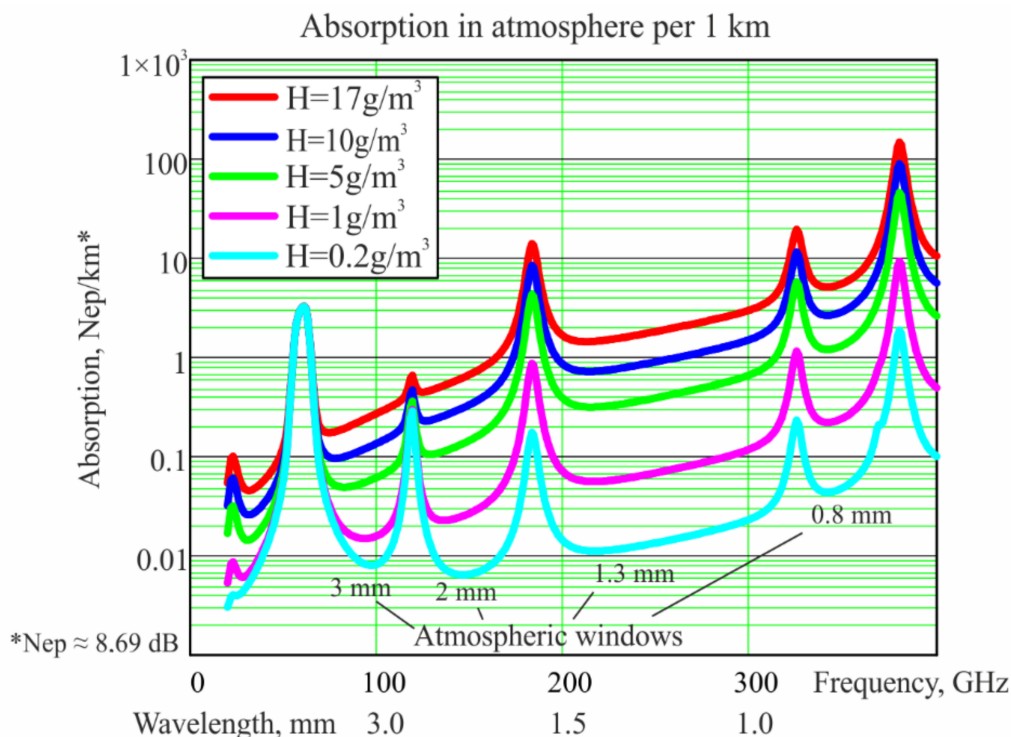

**Figure 4.** A typical picture [23] of atmospheric propagation in the first four subTHz transparency windows (3, 2, 1.3, and 0.8 mm) can be used for open-air terrestrial applications. The transparency windows are separated by intense absorption lines. The peaks are due to absorption lines for oxygen (two leftmost) and water (three following).

### 2.3. Prospects for Gyrotrons on the Suffa Antenna for Radar and Telecommunication Applications

The use of the above-mentioned transmitters, receivers, and antennas under the estimated atmospheric limitations will be assessed. The new instruments allow obtaining data transmission rates for communication and the range and resolution of radars. The content of a radar depends on many factors: the design, signal processing device, operation conditions, etc. Leaving out these conditions, we can estimate the range only roughly, let alone the fact that the basic radar equation itself is approximate, and different test objects under radar surveillance can yield noticeably different results.

Nevertheless, an approximate set of some values will allow a rough estimation of the performance of the system being developed from the following Formula (3) [9]. We can derive the value of the range:

$$r_{\max} = \sqrt[4]{\frac{\gamma E \eta S^2 \sigma}{2\pi N_0 q_{\min}^2 \lambda^2}} \tag{3}$$

where:
$\gamma$ is the coefficient of nonoptimality of the receiving device;
$E$ is the energy of the probing signal;
$E = \tau\, P{\cdot}T$ is the average power, where $\tau$ is the absorption of the atmosphere, $P$ is the transmitter power, and $T$ is the signal duration (signal accumulation time);
$\eta$ is the efficiency of the transmitting antenna;
$S$ is the effective area of the antenna (here, it is assumed that the reception and transmission are carried out by one antenna);
$S = K\, S_A$;
$K$ is the rate of antenna aperture area utilization;

$S_A$ is the geometric area of the aperture for the antennas used,

$$S_A = \pi \left(\frac{b}{2}\right)^2; \tag{4}$$

$b$ is the diameter of the antenna;
$\sigma$ is the effective scattering area of the object, which depends on the shape, orientation, size, and reflective properties of the test object and the wavelength; when probing extended objects, one should take into account the area of reflections determined by the radiation pattern of the radiating antenna;
$N_0 = kT_e n$ is the spectral intensity of white noise;
$k = 1.38 \times 10^{-23}$J/K is the Boltzmann constant;
$T_e$ is the effective ambient noise temperature;
$n$ is the noise factor of the receiver;
$q_{\min}^2$ is the minimum signal-to-noise ratio (minimum in terms of the power at the output of the linear part of the receiver); and
$\lambda$ is the wavelength. Specific numerical values of the parameters of receivers, antennas, transmitters, and atmospheric attenuation used for simulations are presented below.

To estimate the first-approximation maximum range ($r$), radar parameters can be conveniently set as typical, that is, shared by most radars, varying the above-specified parameters of receivers, transmitters, and antennas. It is also convenient to take the generally accepted dimensions of a test object being a metal ball with a diameter of 1 m and $\sigma = 0.8$.

The calculations for $\lambda = 3$ mm made by using Equation (3) without taking into account the influence of the atmosphere give promising results for up to 2500 km. It is more than the thickness of the atmosphere (<100 km). The range of radar without the atmosphere should be much better for 1.3 and 0.8 mm. However, the atmosphere is taken into account in the formula as the additional attenuator $\tau$ along the signal path, the latter being single for a communication channel and double for radar. When a signal is emitted, the atmosphere reduces the effective power of the transmitter. When a signal is received, it reduces both the level of the received signal and adds its own Nyquist noise and other noises to the SNR.

The results for the directions close to the zenith direction for the 2500 m elevation (like Suffa) with the atmosphere yield only 500 km for a 3-mm radar. Only low orbits of space debris and satellites are available. If the antenna is installed at sea level, $r$ becomes about 100 km and makes detection of the object of interest totally impossible. The scenarios for radars at 1.3 mm and 0.8 mm are much worse, and $r$ is less than the thickness of the atmosphere.

A more complex approach can be implemented if we take into account not only the absorption of the atmosphere, but also its scattering and turbulence. It is a main problem of the HUSIR project, and the influence of turbulences should be investigated at specific places of radar antennas. The turbulence patterns are dramatically different in different locations and are the subject of our further research.

For the zenith direction, the solutions are obviously the best, and when the ray is inclined towards the horizon, the range of ($r$) worsens following the cosine function. At an angle of 60 degrees, the thickness of the atmosphere along the beam doubles, as well as the absorption. When the power is attenuated by 1 dB/km (a familiar figure for long subTHz waves at 3 mm), it is not too critical. For shorter subTHz waves (1.3 and 0.8), the figures are between 10 and 100 dB/km. A double passage of a wave through the atmosphere will cause a drop in the received signal power by 100 dB and decrease the range by a factor of 300. These losses are too significant to provide effective communications, astronomical observations, and, in particular, radar operation. Nevertheless, simulations of the atmospheric conditions typical for high mountains (3–5 km) and the best parameters of the contemporary components of radar systems have been performed and are rather promising.

It yields only ~7 km of the beam diameter on the Moon's surface for a 3 mm beam and a spatial resolution of ~5–7 cm for space debris at the ~800 km range.

Communication estimates look more optimistic as the signal has to travel through the atmosphere only once. The equation for assessing the communication performance includes only a square root, unlike the radar's fourth root in Equation (3). The operating range increases essentially; actually, one can speak about any distance.

Everyone knows of the Voyagers [31] as a brilliant example of deep-space communication. Their data transfer can reach extremely far distances, but with very low data transfer rates, practically bit-to-bit transfer. However, Voyagers traveling since 1977 are now at a distance of 20 billion km (~$2 \times 10^{-3}$ light-years), use a frequency of only 8 GHz, and the 3.5 m antenna is still in contact with the Earth.

Let us estimate what messages to extraterrestrial civilizations can be sent using the available gyrotron devices. For example, one can transmit to the Moon at least 500 TV channels simultaneously in a 10 kW CW 3 mm signal and the 4 GHz gain bandwidth. An aggregate of 1000 synchronized gyrotrons with a total power of 2 GW at 160 GHz and a $100 \times 100$ m$^2$ phase array yields the ~$10^8$ gain compared to that of the Voyagers channel.

At this power level, a signal can reach places in the Universe located $4.6 \times 10^{10}$ light-years from Earth, where there are 500 billion galaxies and ~$10^{20}$ stars. Indeed, the SETI (Search for Extraterrestrial Intelligence) [24] program is a rather exotic example demonstrating the possibilities of gyrotrons. However, because of high-speed communication with current Martian and future deep-space missions, we have an excellent reason to use subTHz gyrotrons and low-noise cryogenic receivers to increase rates. A two-order increase in the gyrotron power yields an increase by an order of magnitude in the data transfer rate according to the Shannon–Hartley Formula (4):

$$C = B \, log_2 \, (1 + SNR), \tag{5}$$

where *C* is the bit rate of the channel, *B* is the spectrum bandwidth, and *SNR* is the signal/noise ratio.

With the MW power sources presented here, one can obtain an increase by three orders of magnitude, as compared with those used till now [4] and at the 10 kW level of high-power deep-space station signals. Another order of magnitude, according to Equation (4), can be gained by using the W-band antenna (Suffa, for example) instead of the X-band ones, which are currently used for the Exo-Mars mission [4]. Two more orders of magnitude in *SNR* can be won by using the low-noise superconducting receivers, which are now capable of cryogen cooling to tens of K of the noise temperature compared to the not-cooled receivers used for space communication now. Currently, there are some groups in the USA [36,37] and Europe [23,29,38] who develop and use receivers of the quantum level of sensitivity in combination with actual onboard and terrestrial subTHz antennas. There also exist lots of receiving subTHz systems tested only in laboratory.

## 3. Results of Studying Atmospheric Propagation of subTHz Radiation

Among the surveyed sites, which are noticeably inferior in terms of absorption to the best-known locations for subTHz instruments (Atacama, Tibet, or Pamir) practically everywhere, there are good prospects for working in the first (a wavelength of 3 mm) and second (2 mm) subTHz atmosphere transparency windows. Seasonal variations make it possible to effectively use the instruments installed at these sites and the waves of the third transparency window ($\lambda \sim 1.3$ mm) in winter. Several sites have good prospects even in the summer season; for example, Mount Muus-Khaya in Yakutia is about 2000 m above sea level and located near the Nothern "cold pole" of the Earth. Virtually nowhere among the studied places do the conditions for the effective use of instruments in the studied areas with submillimeter waves (an atmospheric transparency window of 0.8 mm) last long enough to be of any practical use.

Therefore, some conclusions regarding the practical use of the sites under study are negative. In particular, in the vast territory of northeastern Eurasia chosen for experiments,

in contrast to the western hemisphere (Andes, Antarctica, Greenland, etc.) or the more southern places of the eastern hemisphere (Tibet, Pamir, etc.), no place has yet been confirmed to present a good opportunity for sub-mm tools. Even the seasonal options for working in the 1.3 mm transparency window offered by these sites are minimal.

When comparing the main results of the measurements supplemented by the latest data about the studied sites with those previously studied [28], we can see that the most promising site, Muus-Khaya Mountain, is not an excellent choice due to the place being remote and having no infrastructure to be found. Our on-land journey to the site included a 4-day horse ride on the route's final leg, and the return journey was by air onboard a helicopter. The site has no infrastructure or power supply; the only "facility" found there was an old weather station that closed in the 1970s.

An attempt to find a northern option for placing an instrument analogous to the SPO installed near the south cold pole of the Earth in Antarctica was unsuccessful. There are no places above sea level, and the altitudes of about 1500 m in Svalbard are under the influence of the Gulf Stream, which carries a noticeable quantity of moisture even in winter (we carried out a year-round cycle of observations there). Thus, there is no dry air comparable to that of Antarctica.

The conclusions we made after many years of observations on the Suffa Plateau (the site proposed for installation of a 70 m antenna) are not entirely encouraging, not even for the 0.8 mm transparency window. This extreme window of atmospheric transparency, the only one available for this site, shows up for only about a dozen days a year when conditions are favorable for signal propagation. Such fact makes both radio astronomy observations and space communication possible only in the near-zenith directions. As the path gets more inclined to the horizon, atmospheric absorption increases sharply. As for radar applications, the beam has to cross the atmosphere twice, so the problem of locating low-lying objects becomes insoluble.

There are quite promising sites in Siberia (Badary and Mondy) [35], confirmed by satellite data [3]. However, they are rather far from the best record-breaking radio astronomy sites like ALMA and, at the very least, require further research to find the most prosperous locations, which will probably involve some mountain climbing.

The Caucasus seems to be the most attractive in terms of infrastructure, engineering, and staffing, as well as due to the Earth's surface coverage by subTHz instruments [39]. The instrument coverage of the Earth's surface plays an essential role in all three main types of subTHz applications: astronomy, communication, and radar. In astronomy, adequate surface coverage in the UV plane is essential for radio interferometry, which causes great interest in the EHT team working in the transparency window of 1.3 mm in northeastern Eurasia. In a giant interferometer, it is impossible to replace the complete coverage of the surface with a group of telescopes scattered far from one another. In communication and radar, the surface coverage is even more critical. The challenge of keeping in touch with deep-space missions as they disappear under the horizon is obvious. Looking for meteoroids and locating space debris generally requires a total coverage of the Earth's surface by tracking stations. Unfortunately, our last investigations demonstrate that the North Caucasus has not yet revealed any good places for the 0.8 mm window, and the 1.3 mm window appears only for a short time and only in the near-zenith directions. The reason for this is evident once you take a look at the map of rains: the atmosphere is humid due to the proximity of the Black Sea, while the predominant air currents bring in all the moisture from the Mediterranean area, which then condenses on the Caucasus, acting like a giant condenser and making the use of subTHz instruments problematic. Even the ascent to the heights of 2500–4000 m (Arkhyz, Terskol) did not result in any encouraging numbers. In the summer of 2021, a very adventurous attempt was made to instrumentally assess the future use of Mount Elbrus, the highest point in Europe (5642 m), see Figure 5. Long-term monitoring of satellite and meteorological data made it possible to choose the optimal time for ascent and measurements. The experiment was scheduled for the expected window of August 9 to 11. The expedition was successful, unlike the effort of a group of climbers

who followed the same route two weeks later and whose endeavor ended fatally for five of its participants.

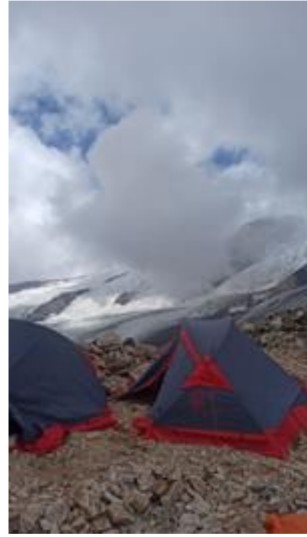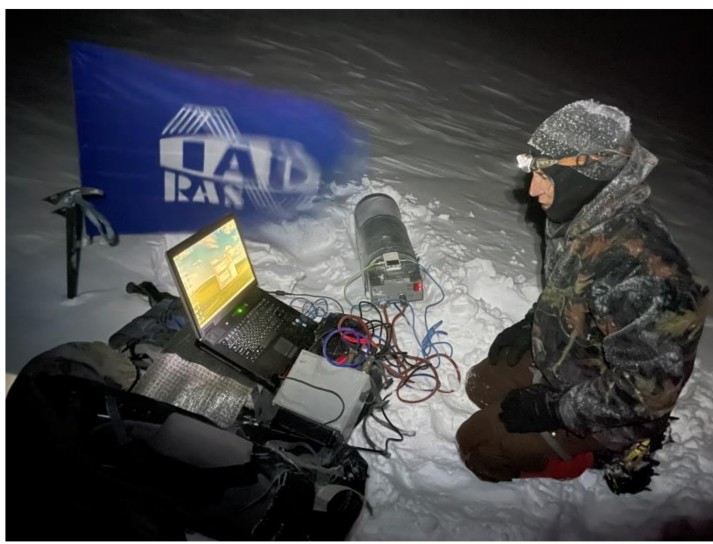

**Figure 5.** Elbrus IAP RAS expedition August 2021: night measurements at 5630 m made by MIAP-2. There is ~0.25 ± 0.25 Np of absorption of 3 mm waves.

However, as soon as the equipment was deployed on the Elbrus slope, the weather worsened sharply, the sky's transparency dropped, and it started to snow. The measurements yielded an atmospheric absorption of ~0.25 Np with an error of ±0.25 Np due to the horrible weather. One might think that this result is of no value at all. However, it has severe scientific significance: the western part of the North Caucasus, including its highest peaks, is too wet and has no prospects for the operation of subTHz instruments due to the limitations of the microwave astroclimate at least up to waves shorter than $\lambda \leq 1$ mm.

Turning the leaf on this list of relatively negative results, we should speak of a number of optimistic and inspiring results for the Year 2021 and further plans. Using unified approaches and uniform methodology and instruments, we explored a vast territory of northeastern Eurasia to map the conditions for the propagation of subTHz radiation in the atmosphere to compare different sites for placing antenna systems for astronomy, telecommunication, and radar applications. Long-term studies on the Suffa Plateau (Figure 6b) have shown good prospects for the completion of construction of the RT-70 antenna and the conditions sufficient to ensure its operability up to wavelengths of 3–2 mm, and revealed, at the same time, that it is unfeasible to erect a costly fine-tuned high-quality 70 m mirror with an operation range of 1.3 and 0.8 mm atmospheric windows due to the astroclimatic limitations of the site.

At the same time, it was proved that it would be much more practical to install a relatively small (15–21 m) high-quality (RMS < 50 μm) mirror, which would ensure efficient operation in the 1.3 and 0.8 mm transparency windows on the slope of the neighboring Aktashtau mountain (3200 m) (Figure 6a) to add it to the main 70 m dish of the Suffa observatory built on the plateau at an altitude of 2400 m. Exploitation of RT-70 at waves shorter than the 3 mm window is quite expensive, requires too much time, and was demonstrated as not effective due to the astroclimate problems at 2400 m. This suggestion was included in the current plans for updating the Suffa project. The selection of a specific site for a smaller mirror requires more detailed studies planned for 2022. Based on the results of direct measurements, some corrections of the efficiency indicators [3] of the Suffa Observatory were proposed as an element of the EHT-derived analysis based on the satellite data. Suffa is much better than the predicted satellite data.

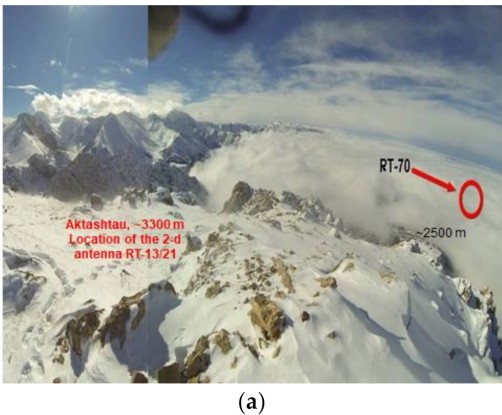
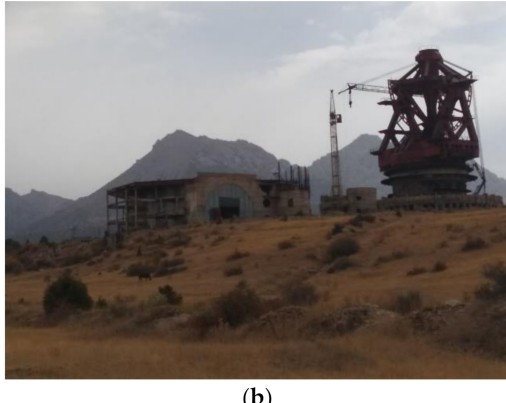

(**a**)                                                      (**b**)

**Figure 6.** Suffa Plateau (2400 m) and Aktashtau mountains (3200 m): views (**a**) from Aktashtau to RT-70 in clouds and (**b**) Aktashtau from the RT-70 location (the left one).

The first results of 2021 from Aktashtau generally confirmed the results previously obtained near Mount Pastukhova (Figure 7) and predicted theoretically based on the atmosphere model. A difference of about 0.01 Np was also observed. However, more thorough and longer-time measurements are required, including the year-round cycle of observations, which has already been completed at three sites (Siberia, Caucasus, and Svalbard) [32]. The Suffa project, as mentioned above, is now viewed as part of the further EHT community [3]. However, predictions of its performance parameters are not quite so good for an element of the EHT, as our estimations have shown.

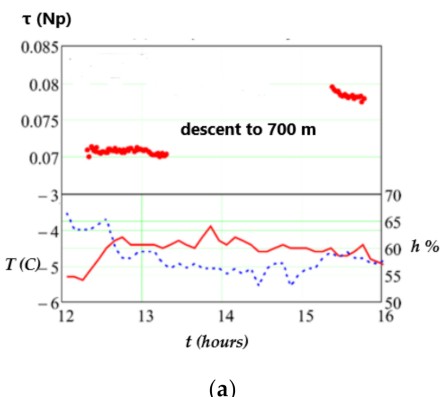
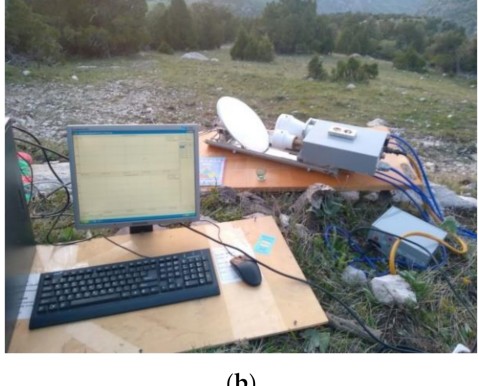

(**a**)                                                      (**b**)

**Figure 7.** (**a**) Difference of $\tau$ (Np) between two points near Mount Pastukhova differing in altitude by 700 m [25]. Measurements of $\tau$ were performed over ~3.5 h at 12:15–15:45 (top picture) at relatively low temperature (red, bottom) and humidity (blue dots) variations. (**b**) Measurements at Aktashtay slope (2900 m) June 2021 by MIAP-2 confirmed possibility to install a small (15–21 m) antenna of the Suffa observatory.

Paper [3] shows that Suffa is a reasonable candidate among the new candidates for EHT elements from the viewpoint of water absorption and a rather suitable one to add to the coverage of the VLBI system. However, Suffa is even better than paper [3] predicts. First of all, contrary to an altitude of 2000 m, indicated in [3], of RT-70 of the Suffa Observatory, the height is almost 2500 m. Hence, at least 0.01 Np of zenith losses can be omitted from the calculation. Another and more important fact is that the second antenna of the Suffa Observatory will be installed not so far from Suffa, at an altitude of ~3000 m, and the gain in the altitude should be taken into account with a decrease of more than 0.1 Np.

As of today, in the western part of the North Caucasus (Karachay-Cherkessia, Kabardino-Balkaria), no good places have been identified as promising for the installation of 1.3 and 0.8 mm instruments [39]. At the same time, based on the preliminary studies and the

study of satellite and meteorological data, there are specific prospects at the sites located to the east: North Ossetia, Chechnya, and Dagestan, where the integral air humidity and precipitation level give substantial grounds to find promising sites for subTHz instruments on the slopes of some mountains at a height of about 3000 m. The meteomap of the Caucasus with a total annual rainfall level indicates the difference in annual rainfalls by more than an order (from 200 mm to 2000 mm), and there are lots of dry places. A preliminary expedition to Chechnya has demonstrated some dry mountains at altitudes of approximately 3000 m. Our first expedition to Chechnya in July 2021, which was undertaken after meteorological and satellite data analysis, confirmed that the east of the Caucasus is much drier than the west. There are some excellent places for a more detailed investigation. The first cycle of direct MIAP-2 measurements in Dagestan and Ossetia in November 2021 [33] gives very promising results (see Figures 8 and 9), and the measurements will proceed in 2022.

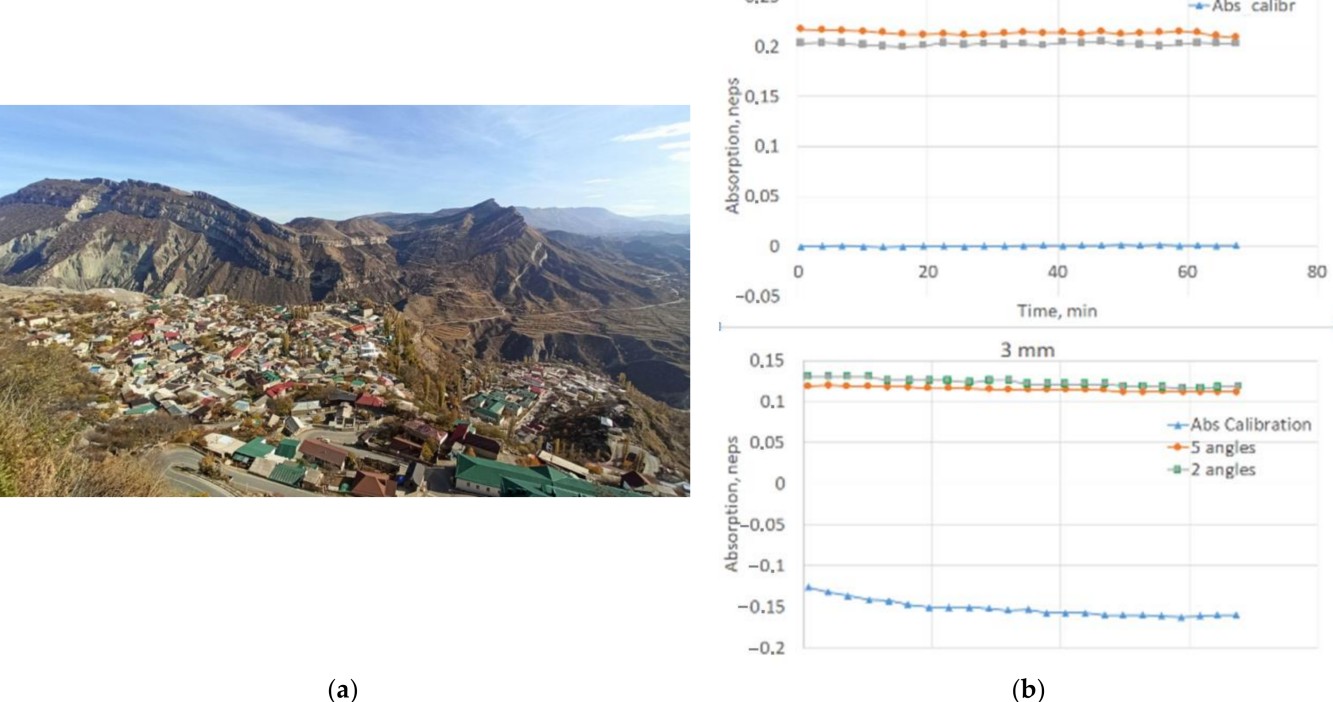

(**a**)　　　　　　　　　　　　　　　　　　(**b**)

**Figure 8.** (**a**) Dagestan, Gunib, Mount Mayak, 2400 m, 350 clear days per year. (**b**) Measurements at $\lambda$ 3 mm and $\tau$ = 0.12 Np and at $\lambda$ 2 mm and $\tau$ = 0.2 Np, 2 November 2021.

Mount Mayak (Figure 8) near Gunib, Dagestan, is a very promising place from the viewpoint of transportation and other infrastructure components. However, it is not high enough, and with $\tau$ and PWV being at a slightly higher optimistic level of 0.1 Np, Mount Shalbuzdag is definitely better (Figure 9).

The hypothesis about the shift of places with the optimal astroclimate in the Caucasus to the east has been confirmed by the expedition presented here. In Dagestan, including several locations at the border with Azerbaijan, there are some more promising high peaks (4–4.5 km), which are drier according to the satellite and GPS climate data, but have not been measured directly to study their astroclimate yet.

However, the conclusion that the Caucasian antenna should be installed in Dagestan or Ossetia is proven here already. The choice of a specific location among the Mayak, Shalbuzdag, and Stolovaya mountains (already surveyed) or Bazarduzu, Khorai, Karakh, or Khunzundalilbak plan to be studied in 2022 will be made later.

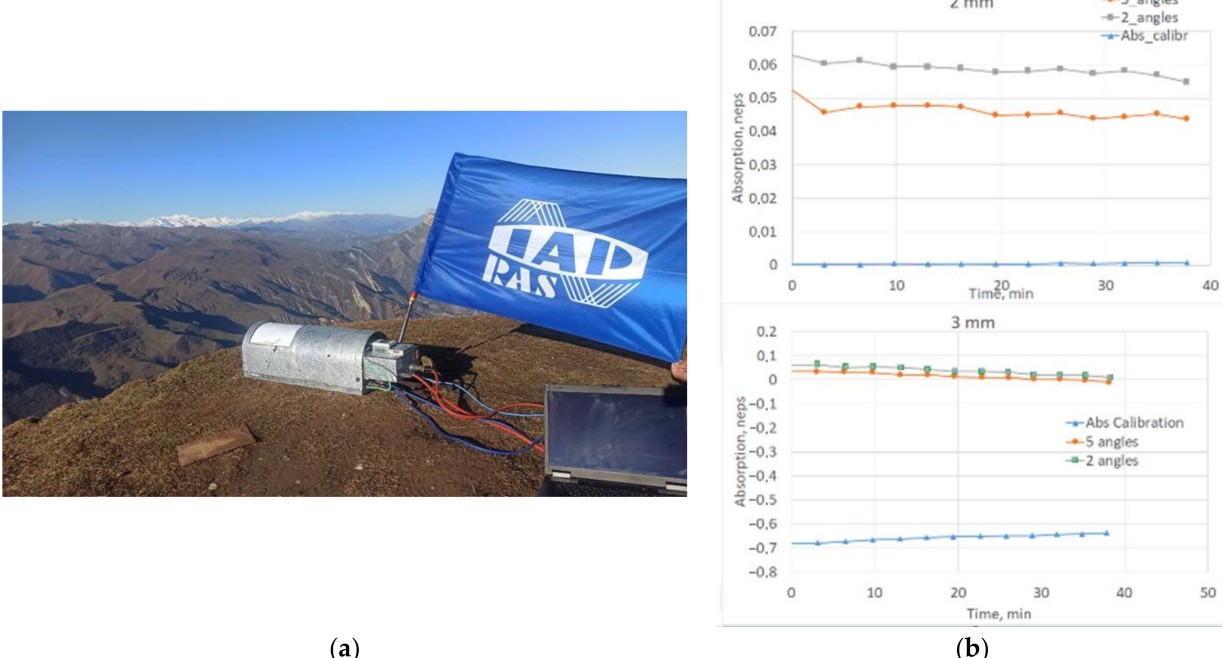

(**a**)                                      (**b**)

**Figure 9.** (**a**) Dagestan, Miskindja, Mount Shalbuzdag, 4142 m. (**b**) Measurements at $\lambda$ 3 mm $\tau$ = 0.07–0.1 Np at 3600 m and at 2900, respectively, at $\lambda$ 2 mm $\tau$ = 0.07–0.12 Np at 3600 m and at 2900, respectively, 7 November 2021. In terms of the water vapor level, it corresponds to a very impressive PWV 2–4 mm.

## 4. Conclusions

The current progress in the development of new subTHz instruments, such as antennas, gyrotrons, and receivers, allows some very ambitious applications to be implemented. In particular, tasks such as mapping the lunar surface or locating space debris can be conducted at a much higher resolution when the instruments are located in places with satisfactory microwave astroclimate, as shown in the present work.

Estimations of possible locations for subTHz instruments in Eurasia made by analysis of space and meteo data have revealed some promising sites, including those that will enhance the EHT coverage. However, all these locations are not as good as the record-breaking radio astronomy sites of the southern hemisphere, and additional studies are required for better estimation of subTHz instrument prospects in northeastern Eurasia. The expeditionary measurements presented here confirmed good prospects for some locations in Dagestan. Based on the performed measurements of atmosphere propagation, corrections to the current plan of the Suffa project have been outlined. It has been proved that it would be much more practical to install a relatively small (15–21 m) high-quality (RMS < 50 μm) mirror, which would ensure efficient operation in the 1.3 and 0.8 mm transparency windows on the slope of the neighboring Aktashtau mountain (3200 m) to add it to the main 70 m antenna of the Suffa observatory built on the plateau at 2400 m. RT-70 should be operated only with cm and mm waves not shorter than 3 mm.

Simulations of the radar and telecommunication systems' performance under atmospheric conditions allow us to adjust existing plans of the new antennas and predict more conclusively the plausibility of such instruments as space debris surveillance radars or deep-space communication hubs, at the same time taking into account the advent of extremely powerful gyrotrons and low-noise superconducting receivers presented in this article.

**Author Contributions:** Conceptualization, M.G.; Data curation, S.S.; Formal analysis, K.M.; Investigation, G.B., A.G., D.D., I.L., A.M. and A.K.; Project administration, G.D.; Supervision, Y.B. and G.S.; Writing—original draft, V.V. All authors have read and agreed to the published version of the manuscript.

**Funding:** This work was supported by the Russian Science Foundation (RSF), Project # 19-79-30071.

**Institutional Review Board Statement:** Not applicable.

**Informed Consent Statement:** Not applicable.

**Data Availability Statement:** Not applicable.

**Acknowledgments:** The authors are very grateful to Mikhail Petelin and Fedor Kovalev for useful discussions. The authors are grateful to the management and staff of the Dagestan and Vladikavkaz scientific centers of RAS, SAO RAS, IAP RAS, Kh. I. Ibragimov Complex Research Institute RAS, Alexeev NNSTU and Lebedev PI RAS for kindly providing the research equipment and assistance in conducting expeditions.

**Conflicts of Interest:** The authors declare no conflict of interest.

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
