# Peer review of "Atmospheric Propagation Studies and Development of New Instrumentation for Astronomy, Radar, and Telecommunication Applications in the Subterahertz Frequency Range"

_applsci, doi:10.3390/app12115670_

Round 1

Reviewer 1 Report

THz waves, including sub-THz waves haved used in the communication, radar. The authors studied the applications of the sub-THz waves in the astronomy, radars, and telecommunications. This topic is very hot in the THz science and technology. However, the present form of the manuscript can be accepted to publish. There are many errors to be revised.

  1. In the Fig. 4, authors should label the frequency position of the four peaks. Note that in this annotation, the wavelength is used. But in the figure, the frequency is used.
  2. In the Section 2, Materials, Methods, Hardware for SubTHz applications. I suggest to add sub-section here.
  3. In the last paragraphs of Page 7, the form and the writting should be rewritted. From " The narrowband tansmission of Voyahers......." to the last lines of this page.
  4.  The title of Section of 3, Results of atmosphere researchers, should be revised.
  5.  I think some figures are necessary in the manuscript. Such as Fig. 5, Fig. 7(b), Fig. 9(a). And the format of the Fig. 8(b) and Fig. 9(b) is needed to revised. The authors just put a picture of the instrument, but the data is not clear. Why not plot the experimental data directly?
  6. In the Conclusion, there are many "Our", generally it is not good as such writing. The authors need to revised these sentences, "Our estimation of possible.....".

Author Response

Point 1:

    In the Fig. 4, authors should label the frequency position of the four peaks. Note that in this annotation, the wavelength is used. But in the figure, the frequency is used.

Response 1: The figure has been corrected, along with the frequency, wavelengths have also been added. The first four subTHz windows of atmospheric transparency are noted, which are of interest for the transmission of astronomical, communication, and radar signals in terrestrial applications. Peaks due to absorption lines for oxygen (two leftmost) and water (three following) are described in the text.

Point 2:     In the Section 2, Materials, Methods, Hardware for SubTHz applications. I suggest to add sub-section here.

Response 2: Done. Thanks!

Point 3     In the Section 2, Materials, Methods, Hardware for SubTHz applications. I suggest to add sub-section here.

    In the last paragraphs of Page 7, the form and the writting should be rewritted. From " The narrowband tansmission of Voyahers......." to the last lines of this page.

Response 3: Rewritten, thanks!

Point 4     The title of Section of 3, Results of atmosphere researchers, should be revised.

Response 4: Reseived. Done.

Point 5:      I think some figures are necessary in the manuscript. Such as Fig. 5, Fig. 7(b), Fig. 9(a). And the format of the Fig. 8(b) and Fig. 9(b) is needed to revised. The authors just put a picture of the instrument, but the data is not clear. Why not plot the experimental data directly?

Response 5: Revised, pictures are replaced with graphics

Point 6    In the Conclusion, there are many "Our", generally it is not good as such writing. The authors need to revised these sentences, "Our estimation of possible.....".

Response 5: corrected

THANKS and sorry!

Reviewer 2 Report

The paper seems to be a review, but after reading, it is just a review of some national activity, as it can be seen in the list of references.

When the authors write

“ Current progress in the development of new sub-terahertz instruments discussed in this paper, including antennas, high power gyrotrons, and low noise receivers, provides a wide range of their possible applications.” They refer mostly to their laboratories not to worlwide activity.

It seems to be more a  paper written for a scientific magazine not for a scientific journal, as it mentions many applications, but not deepening any of them.

The attenuation of the atmosphere considered (see Fig. 4, taken from Liebe) is partial as rainfall/snowfall are not considered.

The details of the instruments shown (e.g., see Fig. 3) are appropriate for a magazine, not for a research journal because no technological details are reported.

Eq.(4),  as Eq.(1) and (2),  are well known. All my students know them. In this paper there is no application of Eq.(4). 

The antenna used (see Eq. at line 237) is a parabola, never mentioned explicitly. Attenuation is usually expressed in dBs not in Nepers.

Author Response

Point 1:

    The paper seems to be a review, but after reading, it is just a review of some national activity, as it can be seen in the list of references..

Response 1: Partially corrected and added to the world state of affairs

Thank you for your comment! Although I can only partly accept it.

The founders and leaders of the direction of powerful subTHz gyro devices are teams from Russia and the USA and have been indicated initially, and their leadership work is noted in the article.

However, the reviewer is right in the fact that teams have appeared in Europe, Japan and China, which are already making quite competitive samples of such equipment. Links to some works by Japanese and German groups have been added. The rest still noticeably lose in terms of characteristics.

The purpose of this work was to evaluate the prospects of applications, if you use the best samples on the best antennas located in the best places.

Point 2:     It seems to be more a  paper written for a scientific magazine not for a scientific journal, as it mentions many applications, but not deepening any of them..

Response 2: Apparently the reviewer is right and a deep consideration of the submitted applications was not the purpose of this work. I wanted to evaluate the prospects and possibilities of using the latest developments in the field of subTHz equipment for these applications. And here, too, the authors limited themselves to only three applications (astronomy, communications, and location), where certain results have already been achieved in the cited scientific papers and the goal was to identify the limiting capabilities of technology for these applications in conditions of a real atmosphere and specific antenna systems. Considering the validity of this remark, we supplemented the introduction with these considerations. Thank you for your comment!

Point 3     The attenuation of the atmosphere considered (see Fig. 4, taken from Liebe) is partial as rainfall/snowfall are not considered    

Response 3:  Yes. This is the base component only, which is superimposed by the influence of clouds and precipitation. In this connection, when choosing the location of the antenna, these factors are also taken into account. The analysis of precipitation presented in the work became a key moment in the transfer of activity from the west of the Caucasus to the east, where the annual precipitation level turned out to be almost an order of magnitude lower (from 2000 mm per year to 200). The cloudless sky for 350 days a year over Gunib, given as a long-term fact, confirms the importance of this parameter. Apparently, we weakly emphasized this result in the paper. Thanks to the reviewer!

At the same time, we cannot fail to note that cloudiness, especially light and small precipitation do not prevent operation on sub-THz waves only introducing some attenuation into the signal, unlike optics. This is the main competitive advantage of subTHz waves compared to optics. But this is rather a response to another reviewer who makes a comparison with optical methods. In this connection, this section has been supplemented in the work. Occasionally (a few percent of the total time) when there is precipitation and very thick clouds, no work is being done.

Point 4     The details of the instruments shown (e.g., see Fig. 3) are appropriate for a magazine, not for a research journal because no technological details are reported.

Response 4: Reseived. Done. The comment is accepted. Thank you! Figures and text are supplemented with a block diagram of the radiometer and technical performances of presented devices.

Point 5:      Eq.(4),  as Eq.(1) and (2),  are well known. All my students know them. In this paper there is no application of Eq.(4).

Response 5: The equation, of course, is applied and references to it in the text are added. Thanks, this was missed initially!

We do not pretend to be insufficiently known for the fundamental equations from Skolnik's textbook of the 60th and Shannon Hartley's basic communication equation. About all our estimates of prospects and limiting possibilities in radar and communication applications are calculated on their basis. And in particular, the conclusions in the last paragraph of section 2, presented after formula 4, were obtained on the basis of an analysis of formula 4. I am very glad that the respected reviewer does not limit himself to teaching students, as is now fashionable, by formulas published during the last five years, forgetting the classics. I do exactly the same and my students clearly remember all these formulas too.

Point 6    The antenna used (see Eq. at line 237) is a parabola, never mentioned explicitly. Attenuation is usually expressed in dBs not in Nepers.

Response 6:  Quite right! The antenna is a Cassegrain structure with a parabolic primary mirror and a hyperbolic secondary mirror.  

A more detailed description of the developed antenna has been added to the section, for which the expected results of using it with the best samples of transceivers have been calculated, provided that it is located in the best places in terms of astroclimate

About Nepers. This fact is clear to me as a radio engineer, and it was difficult for me to get used to the completely different units of measurement used by them in the world of astronomers and people of the atmosphere.

DB are really very good for many applications and are used more often now. We added dB to our description of the picture/.

However, historically, to assess the interaction of electromagnetic radiation with the atmosphere, the term "optical depth" was adopted and was measured in nepers. I note that the classical approach outlined by Liebe, cited No. 1 in our list of references, is presented precisely in nepers. But for the convenience of understanding readers who are unusual for this unit, in parallel with Nep, we added a scale in dB.

I note that it is really possible and quite often used just simply relative units or even %% (for example, the phrase “about half of the radiation at 100 GHz is absorbed in the atmosphere even at zenith angles of passage” is much clearer to the reader than 3 dB and a completely non-integer number of Nep.

And for astronomers who are essentially studying the brightness of astronomical objects and expressing it in Kelvins, the atmosphere in the integral is just an attenuator on the way to the star. And it is most convenient to measure its absorption in the form of the brightness temperature of the sky. If approximately half of the radiation is absorbed, then the atmosphere in the integral gives approximately 130 Kelvin, of which most is the contribution of the lower (first 2-3 km) atmosphere, and above it another 2.7 K of the cosmic microwave background (CMB). And most of the absorption at subTHz is determined by water vapor. In this connection, sometimes the amount of precipitated water PVW calculated from certain measurements in millimeters acts as a unit of measurement. And here it is also clear. 4mm of water overhead is good. 2 mm is excellent, and if more than 10, then bad.

By the way, our device actually directly measures the brightness temperature of the sky and recalculated to Np.

The choice of units of measurement, in our opinion, is a matter of taste of the author and traditions. Here we follow the Liebe tradition. And most of all both of them John Napier and Alexander Bell are Scots but the first one is 3 centuries older :-) He was the first in lagarithms and ln to my mind is better then log. Really both units are permitted

Reviewer 3 Report

The article presents an overview of recent advances in the field of devices operating in the sub-THz frequency range, such as receivers, generators and amplifiers. The authors discuss a range of new applications of great interest enabled by the renewed performance of these devices. One section is devoted to the discussion of sub-THz wave propagation through atmosphere and the selection of optimal locations to place antennas working in this range. Although some of the results may be interesting, for the publication of the manuscript both the methods and the results of the analyses/experiments should be described in more detail, avoiding to address them only in qualitative terms.

Here my comments on the manuscript:

  • The text contains several grammatical errors, starting with misspelled words such as “hogh” (line 37) and “higer” (line 42) up to whole sentences whose meaning is difficult to grasp, such as “There are developed and manactured a line of very powerfull SubTHz generators and amplifiers in collaboration of IAP RAS ang GYCOM” (lines 90-92). Even the title contains a mispelled term (“telecommunicationsin”). Listing all the errors would be impractical, so a general review of the grammatical syntax throughout the text is strongly recommended.
  • Symbols in Equations (1) and (2) should be clarified.
  • Section 2 is entitled “Materials, Methods, Hardware for SubTHz applications. However, materials are not once mentioned.
  • 2 should be announced in the text before showing up.
  • The text must be in English language only. Therefore, Fig.2, Fig.8 (b), Fig.9 (b) and the caption of Fig.1 should be revised.
  • Section 2 is presented as the section “devoted to the progress achieved by the authors” (line 76). However, little space is actually devoted to the results, which are embedded in a longer list of results from other papers. It is sometimes difficult to understand whether the authors are referring to their or others results, as in the sentence “The resulting analysis shows higher than expected atmospheric absorption at this site” (line 191), where it is not immediately clear who carried out the analysis. This problem recurs several times throughout the text.
  • The devices claimed to be presented by the authors are poorly described. For instance, despite the sentence “Performances of presented here SubTHz generators” (line 138), the only information available about such generators throughout the text is that they are “very powerfull” (line 91) and that they work “up to the megawatt level” (line 97). All devices should be described in more detail, supporting the narratives with numbers and graphs.
  • In the literature overview, a focus should be placed on the photonics that has shown remarkable results in the Space context. As example, see these papers, 2017 Flexible photonic payload for broadband telecom satellites: from concepts to system demonstrators Int. Conf. on Space Optics—ICSO 2016 (Biarritz, France) vol 10562 p 105621Y; (2019). Ultra-compact tuneable notch filter using silicon photonic crystal ring resonator. Journal of Lightwave Technology, 37(13), 2970-2980; (2021). Design of a large bandwidth 2× 2 interferometric switching cell based on a sub-wavelength grating. Journal of Optics, 23(8), 085801; (2021, June). Advancement of photonic integration technology for space applications: a x-band scan-on-receive synthetic aperture radar receiver with electro-photonic beamforming and frequency down-conversion capability. In International Conference on Space Optics—ICSO 2020(Vol. 11852, p. 118522W). International Society for Optics and Photonics(2021); Design and Performance Estimation of a Photonic Integrated Beamforming Receiver for Scan-on-Receive Synthetic Aperture Radar. Journal of Lightwave Technology, 39(24), 7588-7599). These are just few examples.
  • Several vague sentences are present such as “Using unified approaches and uniform methodology and instruments,” (lines 393-394), “Long-term studies” (line 397) or “it has been proved that” (line 403). These statements do not allow for a critical assessment of the methods used or of the reliability of the results, so the reader is forced to simply trust the authors' work. All approaches, assumptions and instrumentation used should be explained in more detail.
  • Results are often declined in qualitative terms, which limits their expendability. Expressions such as “The resulting analysis shows higher than expected atmospheric absorption” (line 191), “our experiments show a significant decrease in atmospheric absorption as the altitude rises” (lines 192-193), “Performances […] are high to provide proper technical signal/noise ratio when will be used” (lines 138-140) or “Our first expedition […] confirmed that the east of the Caucasus is much drier than the west” (lines 445-447) should be supported by numerical data and/or graphs, in order to give the reader a clearer picture and provide some practical evidence for the conclusions drawn by the authors.
  • On page 3, three applications are listed as being considered by the authors. However, the third of those applications is not actually developed further.
  • Definition of symbols at line 229 should be clarified better.
  • 8 (b) and Fig.9 (b) should be redone, taking care to use English language only, specifying the axes variables and their measurement units, and possibly putting the original graphs instead of a photograph of them.
  • There are Sections 3 and 5 but there is no Section 4, therefore Section 5 should be renumbered as 4.
  • The acronyms “IAP” (line 75), “SETI” (103) and “VLBI” (line 202) should be made explicit.

Author Response

Point 1:

The article presents an overview of recent advances in the field of devices operating in the sub-THz frequency range, such as receivers, generators and amplifiers. The authors discuss a range of new applications of great interest enabled by the renewed performance of these devices. One section is devoted to the discussion of sub-THz wave propagation through atmosphere and the selection of optimal locations to place antennas working in this range. Although some of the results may be interesting, for the publication of the manuscript both the methods and the results of the analyses/experiments should be described in more detail, avoiding to address them only in qualitative terms.

Response 1: Partially corrected

Point 2    The text contains several grammatical errors, starting with misspelled words such as “hogh” (line 37) and “higer” (line 42) up to whole sentences whose meaning is difficult to grasp, such as “There are developed and manactured a line of very powerfull SubTHz generators and amplifiers in collaboration of IAP RAS ang GYCOM” (lines 90-92). Even the title contains a mispelled term (“telecommunicationsin”). Listing all the errors would be impractical, so a general review of the grammatical syntax throughout the text is strongly recommended.

 Response 2: The authors apologize to the reviewer!

Not being native speakers, when sending the manuscript, we agreed with the editors the opportunity to use the service offered by the editors for grammatical checking of the text and were unacceptably careless with the source text.

Now we have corrected the text to the best of our ability, but we also intend to use the editorial service to align the final version of the manuscript.

Once again, we apologize!

Point 3         Symbols in Equations (1) and (2) should be clarified.

Response 3:  Done, clarified.

.

Point 4     Section 2 is entitled “Materials, Methods, Hardware for SubTHz applications”. However, materials are not once mentioned. 2 should be announced in the text before showing up.

     .

Response 4: Reseived. Materials removed. Done.

Point 5:      The text must be in English language only. Therefore, Fig.2, Fig.8 (b), Fig.9 (b) and the caption of Fig.1 should be revised.

Response 5: Revised. Done

Point 6    Section 2 is presented as the section “devoted to the progress achieved by the authors” (line 76). However, little space is actually devoted to the results, which are embedded in a longer list of results from other papers. It is sometimes difficult to understand whether the authors are referring to their or others results, as in the sentence “The resulting analysis shows higher than expected atmospheric absorption at this site” (line 191), where it is not immediately clear who carried out the analysis. This problem recurs several times throughout the text.

    The devices claimed to be presented by the authors are poorly described. For instance, despite the sentence “Performances of presented here SubTHz generators” (line 138), the only information available about such generators throughout the text is that they are “very powerfull” (line 91) and that they work “up to the megawatt level” (line 97). All devices should be described in more detail, supporting the narratives with numbers and graphs.

Response 6:  Done, we refined the author's results and the results of other teams, performances indicated in digital data.

Point 7 In the literature overview, a focus should be placed on the photonics that has shown remarkable results in the Space context. As example, see these papers, 2017 Flexible photonic payload for broadband telecom satellites: from concepts to system demonstrators Int. Conf. on Space Optics—ICSO 2016 (Biarritz, France) vol 10562 p 105621Y; (2019). Ultra-compact tuneable notch filter using silicon photonic crystal ring resonator. Journal of Lightwave Technology, 37(13), 2970-2980; (2021). Design of a large bandwidth 2× 2 interferometric switching cell based on a sub-wavelength grating. Journal of Optics, 23(8), 085801; (2021, June). Advancement of photonic integration technology for space applications: a x-band scan-on-receive synthetic aperture radar receiver with electro-photonic beamforming and frequency down-conversion capability. In International Conference on Space Optics—ICSO 2020(Vol. 11852, p. 118522W). International Society for Optics and Photonics(2021); Design and Performance Estimation of a Photonic Integrated Beamforming Receiver for Scan-on-Receive Synthetic Aperture Radar. Journal of Lightwave Technology, 39(24), 7588-7599). These are just few examples.

Response 7:  The authors thank the reviewer for a very important omission in the manuscript!

Indeed, optical technologies in at least two of the three proposals we offer have extremely advantageous characteristics. We reviewed the recommended articles and completed the comparisons with the results presented in them and added links.  

Point 8 Several vague sentences are present such as “Using unified approaches and uniform methodology and instruments,” (lines 393-394), “Long-term studies” (line 397) or “it has been proved that” (line 403). These statements do not allow for a critical assessment of the methods used or of the reliability of the results, so the reader is forced to simply trust the authors' work. All approaches, assumptions and instrumentation used should be explained in more detail.

Response 8:  Clarification done see new version of the text

 Point 9    Results are often declined in qualitative terms, which limits their expendability. Expressions such as “The resulting analysis shows higher than expected atmospheric absorption” (line 191), “our experiments show a significant decrease in atmospheric absorption as the altitude rises” (lines 192-193), “Performances […] are high to provide proper technical signal/noise ratio when will be used” (lines 138-140) or “Our first expedition […] confirmed that the east of the Caucasus is much drier than the west” (lines 445-447) should be supported by numerical data and/or graphs, in order to give the reader a clearer picture and provide some practical evidence for the conclusions drawn by the authors.

Response 9:  Corrections and clarification fulfilled, see new version of the text...

Point 10    On page 3, three applications are listed as being considered by the authors. However, the third of those applications is not actually developed further.

Response 10:  The reviewer is right! Communications, radars and astronomy - the areas of study of the team of authors are considered further. The third application presented on page 3 was originally designated as "exotic" and was not considered further, since the authors are not experts here. The corresponding note is inserted into the text

Point 11    Definition of symbols at line 229 should be clarified better.

 Response 11:  Agree. Done. 

Point 12  8 (b) and Fig.9 (b) should be redone, taking care to use English language only, specifying the axes variables and their measurement units, and possibly putting the original graphs instead of a photograph of them.

Response 12:  The original screen shots of measurements taken directly on the expedition have been replaced with correct graphs

Point 13    There are Sections 3 and 5 but there is no Section 4, therefore Section 5 should be renumbered as 4

Response 13:  You are right . Corrected. Abbreviations expanded when they first appear in the text.

Thank you!

Round 2

Reviewer 1 Report

now it can be published by Applied Sciences.

Author Response

English and style have been revised

Reviewer 2 Report

The paper has been sufficiently improved to warrant its publication as is.

Author Response

No questions

No comments

Thanks!

Reviewer 3 Report

The Authors replied concisely to the Reviewer questions. The suggestion of the publication can be carried out only after an extensive editing of the English.

The manuscript cannot be published as it is.

Author Response

Extenive editing made by the expert

This manuscript is a resubmission of an earlier submission. The following is a list of the peer review reports and author responses from that submission.

Round 1

Reviewer 1 Report

The manuscript contains many typos or wrong use of English words. It does not define a lot of acronyms, which is a basic requirement  for scientific writing. Most of all, it lacks originality/novelty. Everybody knows water vapor is going to be a huge problem for sub-millimeter radars, and you probably can get a pretty good estimate of its effects just based on meteorological measurements from satellite and/or in-situ measurements such as radiosondes, as the authors have pointed out. So what's the significance of your work? Apparently daily/hourly variations of water vapor is expected at any locations, but a climatological value is probably good enough for you to select a site for your radars?

Following are just some example of typos/errors in the paper:

(1) Line 104: This should be Fig.1, not Fig.2. Also, it should be "about 100 microseconds", not "abort 100 microseconds".

(2) Line 119: should it be "atmospheric conditions", not "conclusions".

(3) Line 139-154: This paragraph is the same as the paragraph from line 106-121.

(4) Line 187, change "water and vapor" to "water vapor".

and many more.....

Reviewer 2 Report

The paper is very hard to read. Review of literature (very scarse, the authors neglect a large amount of work done at the frequencies of their interest concerning radio propagation, and theoretical studies - see IEEE, MDPI, Wiley journals) , of technology, of purpose of the paper and partial conclusions are mixed together making the paper not understandable. The results are very few and not sufficiently mature to be published. Classic formula, such as (4) are reported with no application with data, etc. The paper seems to be written, badly, for some students.

Reviewer 3 Report

Dear authors

We have carefully reviewed your manuscript titled “Influence of atmospheric conditions on the propagation of a wave beam in the subterahertz frequency range”. The authors first discussed current progress in the development of new sub-terahertz instruments, then presented some valuable results of atmospheric properties conducted with unique hardware, using their original method of direct measuring, calculated radar and telecommunication systems with new components operating, finally gave a new outlook for developing new antennas at Suffa Plateau and in the Caucasus and possible applications of new instruments based on extremely powerful gyrotrons and low noise superconducting receivers. The results are of importance to the development and applications on sub-terahertz. This referee suggests a major revision of the text to make it suitable for publication.

Comment (1) In introduction, it is better to add specific astronomical applications of subterahertz technology. Pulsars are famous as highly magnetized and rapidly rotating neutron stars providing us with an opportunity to explore physics under extreme conditions (e.g., Gao et al. 2017, ApJ, 849,19; Wang, et al. Universe, 2020,6,63; Deng et al. 2021, ApJ, 909,174). More than 50 years of observations have shown that, pulsars emit radiation in multiple wavebands, including radio, infrared, ultraviolet and visible light, X rays, and gamma rays, therefore, astronomers have set up a large number of detectors on the ground and in space, including subterahertz receivers, to study the motion and evolution of pulsars. The subterahertz techologies are also widely used in the study of star formation, galaxy evolution, dark matter and cosmology (e.g., Gao et al. 2019, Astron. Nachr. 340,241).

Comment2) From the title of the article, this paper should be an experiment-led research article. However, the core content of the paper is too little, this referee strongly suggest the authors to add necessary measurements, equipment parameters, and statistical climatic conditions. Given the influence of atmospheric conditions on the propagation of a wave beam in the subterahertz frequency range, the authors should further discuss other properties of subterahertz pulses, e. g., the characteristics of time domain and spectrum variation of subterahertz pulses, and its dependence on water vapor concentration and transmission distance and so on.

Comment3) Throughout the text, the formulas and their explanations are not rigorous enough, and the writing style including language is not standardized enough.

The above changes are recommended before this article is published.

Best wishes

Referee
